# Effect of Germination on Alfalfa and Buckwheat: Phytochemical Profiling by UHPLC-ESI-QTOF-MS/MS, Bioactive Compounds, and In-Vitro Studies of Their Diabetes and Obesity-Related Functions

**DOI:** 10.3390/antiox10101613

**Published:** 2021-10-13

**Authors:** Simon-Okomo Aloo, Fred-Kwame Ofosu, Deog-Hwan Oh

**Affiliations:** Department of Food Science and Biotechnology, College of Agriculture and Life Sciences, Kangwon National University, Chuncheon 24341, Gangwon-do, Korea; okomosimon@gmail.com (S.-O.A.); fkofosu17@gmail.com (F.-K.O.)

**Keywords:** metabolites, seed, sprout, antioxidant, enzyme inhibition, UHPLC-ESI-QTOF-MS/MS

## Abstract

Germination can be used to enhance nutritional value and health functions of edible seeds. Sprouts are considered healthier than raw seeds because they are richer in the basic nutritional components (carbohydrates, protein, vitamins, and minerals) and also contain more bioactive components responsible for various biological activities. The effect of sprouting on the antioxidant, antidiabetic, antiobesity activities, and metabolite profiles of alfalfa and buckwheat seeds was investigated in this study. DPPH radical scavenging activity was highest in buckwheat sprouts followed by alfalfa sprout, buckwheat seed, and alfalfa seed, respectively. ABTS radical scavenging potential showed a similar trend as DPPH with buckwheat sprouts exerting the best scavenging capacity. Alfalfa sprout and buckwheat seed exhibited the highest percentage inhibitory activity of α-glucosidase (96.6 and 96.5%, respectively). Alfalfa sprouts demonstrated the strongest inhibitory activity against pancreatic lipase (57.12%) while alfalfa seed showed the highest advanced glycation end products (AGEs) formation inhibitory potential (28.7%). Moreover, thirty-three (33) metabolites were characterized in the seed and sprout samples. Sprouts demonstrated a higher level of metabolites compared to raw seeds. Hence, depending on the type of seed and the target activity, sprouting is a good technique to alter the secondary metabolites and functional properties of edible seeds.

## 1. Introduction

In the past, human beings have been dependent on plants as sources of medicine as well as foods. Recently, due to changing lifestyles, unhealthy eating habits, and increasingly stressful daily life, the global prevalence of various diseases and chronic disorders, including diabetes and obesity, have been on a gradual rise [1]. Despite the considerable progress in the development of possible therapies, the treatment of diabetes and obesity with synthetic drugs has not been fully successful. This is especially true due to the many side effects of conventional diabetes drugs [1].

The unfavorable outcomes related to oxidative stress, the growing number of diabetic patients, and increasing obesity-related side effects have led to the increasing demand to find remedies for the conditions. The metabolic enzymes such as α-glucosidase and pancreatic lipase are important in diabetes and obesity control since they are involved in hydrolyzing activities that elevate blood glucose levels and fat accumulation in the body, respectively [2]. Furthermore, preventing the likelihood of glycation of macromolecules implicated with obese and diabetic conditions is considered key to managing these conditions [3]. Plant-based foods and ingredients are considered natural harbor for a wide range of health-beneficial compounds that, when utilized, may offer health benefits beyond nutritional requirements. Nowadays, the quest to find treatment for diabetes and obesity by developing natural antidiabetic and antiobesity compounds from plant-derived products is shaping the management of these conditions.

Recently, due to their perceived health benefits, the global consumption of sprouts has been on the rise. Sprouting is an inexpensive and effective method to improve the nutritional quality and the phytochemical profiles of edible seeds [4]. The impact of germination on the functional properties of alfalfa and buckwheat has been investigated [4,5]. Owing to their numerous health benefits, an ongoing study on the sprouts is crucial to fully uncover the chemical composition and health functions of these products. Untargeted metabolites analysis based on ultra-high performance liquid chromatography (UHPLC) coupled with mass spectrometry (MS) and the hybrid mass analyzer quadrupole-time-of-flight (QTOF) offers an excellent mass accuracy and measures true fragmentation pattern of unknown metabolites, making the identification of compounds easy and efficient. The objective of this study was to investigate the effects of sprouting on the metabolite profiles of alfalfa and buckwheat seeds using UHPLC-Q-TOF-MS/MS^2^ based metabolite profiling, their antioxidant, antidiabetic, and antiobesity capacities in-vitro. The study is significant for uncovering the impact of germination on the functional properties of buckwheat and alfalfa seeds.

## 2. Materials and Methods

### 2.1. Plant Materials Collection

Alfalfa and buckwheat seed and sprouts were obtained from Charm-Chae-One, Ltd. (Jincheon, Chungbuk, Korea) (Figure 1). According to the vendor, the sprouts were 5 days old, germinated in the dark at temperature (28–30 °C). The samples were dried in an oven, pulverized into a fine powder, extracted, and concentrated. The final supernatant was freeze-dried to obtain crude extract used in the study.

### 2.2. Chemicals

The 2,2-diphenyl-1-picrylhydrazyl (DPPH), 2,2-azino-bis (3-ethylbenzothiazoline-6-sulfonic acid (ABTS), enzymes, substrates, and standards used in this study were purchased from Sigma-Aldrich (Seoul, Korea). The reagents, 4-nitrophenyl α-d-glucopyranoside, sulphuric acid, potassium phosphate mono, and dibasic, sodium dihydrogen phosphate, sodium chloride, sodium citrate, vanillin, sodium carbonate, Folin–Ciocalteu reagent, and potassium persulfate, methyl cellosolve were purchased from Sigma-Aldrich (Seoul, Korea). All chemicals were of the analytical grade.

### 2.3. Methods

#### 2.3.1. Preparation of Ethanolic Extracts

The ethanol extracts were prepared following the procedures previously described [6] with few changes. Briefly, 5 g of each sample was weighed into 70% ethanol (1:20 *w*/*v*) and extracted in an orbital shaker at 50 °C for 1 h. The extracted sample was centrifuged at 4000× *g* for 10 min and the supernatant was collected. The procedure was repeated twice under the same conditions. The final supernatant of each sample was evaporated under vacuum at 40 °C and then freeze-dried. The freeze-dried products were reconstituted in ethanol for further analysis.

#### 2.3.2. Total Phenolic Content (TPC)

Total phenolic content (TPC) of ethanolic extracts was assayed following previous methods using Folin–Ciocalteu [7]. The absorbance of the reactants was read at 765 nm using SpectraMax i3 plate reader (Molecular Devices Korea, LLC, Seoul, Korea). The blank was performed alongside to correct the errors that may have resulted from color interference. The TPC was expressed as milligram (mg) of ferulic acid equivalents per 100 g of sample based on a standard curve.

#### 2.3.3. Total Flavonoid Content (TFC)

The total flavonoid content (TFC) of ethanol extracts was determined using the colorimetric methods as previously performed in our laboratory [8]. In brief, 250 µL of 1 mg/mL of the sample extracts or the standard was pipetted into the wells of microplates. An amount of 75 µL NaNO2 (50 g L^−1^) and 1 mL distilled water were added, and the mixture was allowed to settle for 5 min. An amount of 75 µL AlCl_3_ (100 g L^−1^) was then added, allowed to settle for 6 min followed by the addition of 500 µL of 1 M NaOH and 600 µL distilled water. The TFC was calculated from the catechin standard curve after reading the sample absorbance at 510 nm using SpectraMax i3 plate reader and expressed in milligram catechin equivalents per 100 g of sample (mg CE/100 g).

#### 2.3.4. Total Saponin Content (TSC)

The total saponin content (TSC) was determined according to the earlier described procedure [9] using 72% sulfuric acid and 8% vanillin dissolved in ethanol. The reaction mixture was incubated for 20 min at 60 °C, cooled, and the absorbance was taken at 544 nm using SpectraMax i3 plate reader. The total saponin content was calculated from the soy saponin B standard curve and expressed in milligram soy saponin B equivalents per 100 g.

#### 2.3.5. DPPH Radical Scavenging Activity

The DPPH radical scavenging activity of the extracts was performed following procedures earlier described [10] in 24-well microplate using DPPH solution (4 mg DPPH in 100 mL 95% *v/v* methanol) and Trolox as standard. The absorbance was measured at 517 nm using a microplate reader against ethanol as a blank. The DPPH scavenging activity of the extracts was expressed in micromole Trolox equivalent/gram, dry weight (µmol TE/g, DW) from Trolox standard

#### 2.3.6. ABTS Radical Scavenging Activity

The ABTS radical scavenging assay was based on the reduction in the green ABTS radical cation [11]. An equal volume of ABTS radicle and potassium persulphate solution (2.45 mmol/L) was mixed and incubated for 16 h at room temperature in the dark. During the analysis, the ABTS•+ solution was diluted with ethanol to obtain an absorbance of 0.70 at 734 nm. Following, 80 µL sample extracts or the standard (1 mg/mL) was added to 1 mL of the freshly prepared ABTS+ solution, and absorbance was measured at 734 nm using a microplate reader against ethanol as a blank. The ABTS radical scavenging activity of the tested samples using a similar method described for the DPPH assay.

#### 2.3.7. Pancreatic Lipase Inhibition Assay

The assay for the lipase test has been outlined [8]. 4-MU oleate dissolved in methyl cellosolve was used as a substrate. An amount of 50 µL (50 U/mL) of lipase enzyme was dissolved in methyl cellosolve and mixed with 50 µL of the sample extracts (1 mg/mL). Orlistat was used as the standard. After settling for 10 min at room temperature, 100 µL of 1 mM 4-MU solution was added and incubated for 30 min at room temperature. An amount of 100 µL of 0.1 M, pH 4.2 sodium citrate solution was added to stop the reaction. The fluorescence of the samples was read using a microplate reader at wavelengths of 355 nm and 460 nm. The percentage lipase inhibitory activity of the extract was calculated as shown below:$$\mathrm{Lipase}\;\mathrm{Inhibition}\;(\%)=\left[1-\left(\frac{Ftest-Ftest\;balnk}{Fcontrol-Fcontrol\;blank}\right)\right]\times 100\%$$
where *Ftest*: fluorescent readings of the test samples or the standard with the substrate 4-MU oleate; *Ftest blank*: fluorescent readings of the test samples or standard without the substrate 4-MU oleate; *Fcontrol*: fluorescent readings of the control with the substrate 4-MU oleate; *Fcontrol blank*: fluorescent readings of the control without the substrate 4-MU oleate.

#### 2.3.8. α-Glucosidase Inhibitory Assay

Previous methods were adopted to assess the α-glucosidase inhibitory activity of the extracts [12]. Briefly, 100 µL extract or acarbose (1 mg/mL) was pipetted and was mixed with 100 µL of freshly prepared a-glucosidase (0.5 U/mL). An amount of 300 µL of 10 mM potassium phosphate buffer (pH 6.8) was added. The reaction mixture was incubated at 37 °C for 15 min before proceeding to the next stage. After the 15 min pre-incubation, 100 µL of 5 mM p-nitrophenol-α-d-glucopyranoside substrate was added and the final reaction mixture was incubated at 37 °C for a further 15 min. An amount of 400 µL of the stop solution (200 mM sodium carbonate) was added and the absorbance reading was taken at 405 nm using a SpectraMax i3 plate reader (Molecular Devices Korea, LLC, Seoul, Korea). The sample blanks containing the test sample, substrate, and buffer, but without α-glucosidase, were also assayed. The percentage inhibition of α-glucosidase was calculated according to the formula.
$$\mathrm{Inhibition}\;(\%)=\left(\frac{Ab-Aa}{Ab}\right)\times 100\%$$
where *Aa* and *Ab* are the absorbance values for the extracts (or standard) and blank sample, respectively.

#### 2.3.9. Inhibition of AGEs Formation

Advanced glycation end-products (AGEs) inhibition assay was carried out based on the reaction between bovine serum albumin (BSA) and D-glucose [12] with few modifications. Equal volume (333 µL) of Bovine serum albumin (5.0 mg/mL), D-glucose (36 mg/mL), and negative control or the test samples or aminoguanidine used as a positive control (concentrations 1.0 mg/mL) were mixed in Eppendorf tubes. The mixtures were incubated at 37 °C for a week. Fluorescent AGEs of the mixtures were monitored on a microplate reader using 340 and 420 nm as the excitation and emission wavelengths, respectively. Experiments were carried out in duplicate, and the percentage of the AGE inhibition was obtained as follows:$$\mathrm{Inhibition}\;(\%)=\left[1-\left(\frac{Fluorescent\;of\;the\;test}{Fluorescent\;of\;control}\right)\right]\times 100\%$$

The results were calculated and expressed as percentage (%) inhibition of AGEs formation by the ethanol extracts (1 mg/mL).

#### 2.3.10. UHPLC-Q-TOF-MS/MS^2^ Phenolic Compounds Identification

The bioactive metabolites of the ethanol extracts were analyzed by UHPLC (SCIEX ExionLC AD system, Framingham, MA, USA) following the procedures already described in the previous work performed in our laboratory [8]. The analytical column comprising 100 × 3 mm, Accucore C18 Column was used. An amount of 10 µL of the sample was injected into the system using an autosampler and eluted into the column using a binary mobile phase made up of various constituents denoted as A and B containing 0.1% formic acid in water and methanol, respectively. The Q-TOF-MS was set for the negative mode in a mass range of 100–1000 and a resolution of 5000. The capillary voltage was 1.45 kV while cone voltages used was 30 V. The flow rate of Helium (gas in the cone) was 45 L/h, and the flow rate of desolvation gas (nitrogen gas, N_2_) was 900 L/h. The temperature of nitrogen gas was about 250 °C and the ion source temperature was 120 °C. The collision energies needed to record the MS/MS spectra were established at 15, 20, and 30 V. The system was set to run at a 0.4 mL/min flow rate during the analysis.

#### 2.3.11. Statistical Analysis

The data analytical procedure was performed using GraphPad Prism 8.0 (GraphPad Software, San Diego, CA, USA). The differences of mean values of the various activities alfalfa and buckwheat extracts were determined using one-way analysis of variance (ANOVA) and Tukey’s test at *p* < 0.05 significance level. The results of the analysis were presented as mean ± standard deviation (SD). The principal component analysis (PCA) and heat map were performed using OriginPro 2021 and ClustVis (https://biit.cs.ut.ee/clustvis), respectively (accessed on 20 July 2021).

## 3. Results

### 3.1. Metabolites Identification

Plant metabolites are recognized as important phytotherapeutics with a variety of biological activities in the prevention of a wide range of chronic diseases. In the present work, the effect of sprouting on the metabolite profiles of buckwheat and alfalfa seeds was investigated using the UHPLC-Q-TOF-MS^2^ based metabolite profiling technique. The spectral data obtained were used to tentatively identify the metabolites in seeds and sprouts. The negative and positive ion models were compared for their strengths in identifying the metabolites. The two models presented almost similar metabolites. However, the negative ion model presented more metabolites, hence it was applied for the final identification of the compounds presented in the study. The metabolites were identified based on their potent antioxidant, antiobesity, and antidiabetic properties. The metabolite tentative identification was achieved by comparing spectral data with those available in authentic database standards, Metlin (https://metlin.scripps.edu) (accessed on 20 July 2021) and Metabolomics Workbench (https://www.metabolomicsworkbench) (accessed on 20 July 2021), and cross-checked with related literature reports. The mass spectra data are presented in Table 1 and Table 2. In sprout extracts, a total of thirty (30) metabolites were identified while fifteen (15) metabolites were characterized in the raw seed extracts. Generally, sprout extracts showed a higher metabolite profile compared to seeds. This indicated that the sprouting process led to enhanced metabolites in the seeds.

### 3.2. Heat Map and Principal Component Analysis (PCA)

The principal component analysis was used to discriminate between the tested samples based on their metabolite profiles. Figure 2B–D shows the PCA outcome. Moreover, the data in Appendix A was used to develop a heat map to show the changes in metabolites in the seeds and sprouts (Figure 2A). Red color shows the metabolites with the high concentration while blue color indicates compounds with low concentration. Accordingly, the heat map analysis scored all the extracts obtained from germinated seeds as superior in metabolite profile compared to raw seeds.

### 3.3. TPC, TFC, and TSC and Antioxidant Potential of Ethanol Extracts

The result of total phenolic, flavonoid, saponin contents, and antioxidant potential of the ethanol, were represented according to Table 3 (below). The TPC, TFC, and TSC were measured in terms of ferulic acid, catechin, and soy saponin B equivalent, respectively. Total antioxidant content for the extracts was expressed in terms of micromole Trolox equivalent. TFC, TPC, and TFC of the ethanol extracts of alfalfa and buckwheat were differently affected by the germination process. Generally, germination improved the total antioxidant of the seeds. Thus, the germination process was effective in enhancing the health function with respect to antioxidant activity of the seeds.

### 3.4. Antidiabetic and Antiobesity Activities In Vitro

#### Enzymes and Advanced Glycation End Products Formation Inhibitory Activities

The percentage of antidiabetic and antiobesity properties as measured by enzymes and products formation inhibitory activities (AGEs) are shown in Figure 3A–C. The germination process improved alpha glucosidase and lipase activities of alfalfa but reduced their AGEs activity. On the other hand, the process reduced the alpha glucosidase activities of buckwheat seeds but did not significantly affect their lipase activities.

## 4. Discussion

Table 1 represents the details of metabolites identified in seed extracts with their typical fragments (*m*/*z*). Phenolic compounds were significantly characterized in the seed extracts. Compound **4** was tentatively identified as fumaric acid by comparing its *m*/*z* with those available in the Metlin database. Fumaric acid (FA) is a natural organic acid and a key intermediate of the citric acid cycle. Gülçin et al. reported on the antioxidant activity of FA alongside other phenolic acids [13]. Fumaric acid ester has also been shown to promote the gene factor which regulates antioxidant pathways [14]. Compounds **13** and **15** showed deprotonated molecule [M−H]—at *m*/*z* 289.07 and 441.08, respectively. The compounds, 13 and 15 were polyphenols subclass, flavones identified as (+)-Epicatechin and (−)-Catechin gallate, respectively, after comparing MS data with authentic databases and cross checking with the available literature evidence [15]. In the present study, (+)-Epicatechin and (−)-Catechin gallate were higher in buckwheat seed by 2- and 5-folds, respectively, compared to their sprout. These observations agree with those described by Beitane et al. who noted a significant reduction in catechin levels from 39.13 to 14.14 mg 100 g^−1^ DW upon germination of buckwheat seeds [16]. Moreover, compound **14**, actinonin, was found to be abundant in seeds. Even though it has not been shown that actinonin exerts any antidiabetic, antiobesity, or antioxidant, its health benefit in cancer management has been revealed [17]. Among the amino acids identified in raw buckwheat and alfalfa extracts, there were six (6) (L-Arginine, L-Asparagine, L-Glutamate, Pyroglutamic acid, and N-Methylglutamic acid).

In Table 2 (above), the identification of metabolites in sprout extracts was achieved using similar methods described for the seed extracts. Amino acids were the majority of the metabolites identified in the sprouts. In total, sixteen (16) amino acids were identified in the sprout extracts. Essential amino acids identified in sprouts included lysine, L-Arginine, L-Histidine, L-Valine, leucine, L-Phenylalanine, and L-Tryptophan. L-Phenylalanine and leucine were however, only present in alfalfa sprouts. Non-essential amino acids identified were asparagine and L-glutamate. Emerging evidence suggests that amino acids are beneficial in the prevention of diabetes and related complications. These compounds can either control glycemia and glucose-triggered pathological pathways or reduce oxidative stress and glycoxidation generation [18]. Glycine and lysine are important in the prevention of glycation while phenylalanine and other branched-chain amino acids improve insulin sensitivity and postprandial glucose disposal [18]. Furthermore, in industrial application, lysine-containing peptides can substantially retard the browning reaction with glucose, hence foods with high lysine levels are suitable for lysine fortification of sugar-containing foods which are heat treated. Derivatives of amino acids such as DL-Homoserine, glutathione, and DL-o-Tyrosine were also found in abundance in the sprouts. Glutathione is derived from the amino acids glycine, cysteine, and glutamic acid, and is considered as the body’s super antioxidant [19] (Giblin, 2000). γ-Aminobutryic acid (GABA) was identified at deprotonated molecule [M−H]—(*m*/*z*), 102.05. GABA is a non-protein amino acid with known health benefits including its role as antioxidant and antidiabetic metabolites [20].

Phenolic compounds were substantially present in sprouts. Catechin derivatives were detected in sprouts but in lower levels compared to seeds. Compound **24** displayed deprotonated molecule [M−H]—at *m*/*z* 147.04 and was identified as a trans-Cinnamic acid. It has been demonstrated that trans-Cinnamic acid is derived from the deamination of L-phenylalanine, a reaction catalyzed by L-phenylalanine ammonia-lyase (PAL) [21]. The synthesis of trans-Cinnamic acid in alfalfa sprouts may have been due to the activation of the PAL enzyme by environmental stresses. The antioxidant activity of the compound was described [21]. Compound **26** showed deprotonated molecule [M−H]—at *m*/*z* 163.04 with ion fragment patterns, 163, 91, and the most intense being 119, and it was identified as m-Coumaric acid in buckwheat sprouts. The antioxidant potential of m-Coumaric has been reported [22].

The results obtained in the present investigation also indicate that sprouts are a good dietary source for natural organic acids. Compound **9** showed deprotonated molecule M−H-at *m*/*z* 195.05 and ion fragmentation patterns of 75, 59, 85, 129, and 99, and compound 10 had an ion fragmentation pattern of 71, 57, 85, and 129. Compounds **9** and **10** were identified as gluconic and galactaric acid, respectively. Gluconic and galactaric were not previously present in the raw seeds but were synthesized during alfalfa seed germination. The levels of compound 14, tentatively identified as malic acid, were less in alfalfa sprouts compared to the raw seeds. Moreover, compound 21, tentatively identified as citric acid, was enhanced by 8fold during alfalfa seed germination but was reduced by 1-fold in buckwheat sprouts (Appendix A). L-ascorbic acid was not present in raw alfalfa seeds but was synthesized by the germination process. A similar observation was reported by Wang and colleagues who described a 22.1-fold increase in vitamin C levels in germinated flaxseed [23]. The pathway for L-ascorbic acid biosynthesis has been discussed [24]. L-ascorbic acid is an essential metabolite for different physiological activities in the plant and humans [23]. Due to their abundance of organic acids, the current findings hint at the potential of using sprout extracts as alternative food preservatives in the food industry to enhance the safety of foods.

In Figure 2B, along the PCA axis, the alfalfa sprouts were discriminated from the alfalfa raw seeds based on their metabolite profiles. Alfalfa sprouts recorded a higher metabolite profile compared to the raw seeds. Figure 2C demonstrated that the buckwheat sprouts had better metabolite levels than the seeds. Citric acid, 1-Hexadecylamine, adenosine, and 3-Furoic acid were the most abundant metabolites in raw alfalfa while gluconic acid, L-Phenylalanine, L-Histidine, L-glutamate, D-ornithine, and DL-Homoserine were the most profound compounds in alfalfa sprouts which could be used as biomarkers for the sprouts and seeds, respectively. Moreover, citric acid concentration was the highest metabolite in buckwheat sprouts followed by 1-Hexadecylamine whereas gluconic acid showed the most intense level in the raw buckwheat seeds. In Figure 2D, the combined PCA biplot revealed that alfalfa sprouts had the highest metabolite profile while raw buckwheat seeds had the least concentration of metabolites.

### Enzymatic Bioconversion of Compounds during the Germination Process Led to the Synthesis of Potential Diabetes Biomarkers

This section summarized examples of the newly synthesized metabolites which could be possible biomarkers for diabetes by enzymatic bioconversion of certain compounds. The processes that may have led to their synthesis after germination were described based on evidence from the literature. Figure 4A–C illustrates the peak levels of these compounds and their raw materials. Enzymatic synthesis of GABA involves the glutamate decarboxylase (GAD) enzyme which catalyzes the bioconversion of the L-Glutamate, a predominant raw material for GABA formation [25]. Comparably, in the current findings for GABA, the levels of its raw material (L-Glutamate) were decreased by the germination process while GABA synthesis was observed in germinated buckwheat. Seedlings produce antioxidant compounds such as GABA to cope with environmental stress during germination [26]. Enzymes such as glutamate decarboxylase (GAD) can be triggered to catalyze the alpha-decarboxylation of L-Glutamate to form GABA during sprouting [26]. Therefore, it is apparent that as germination progresses, some of the L-Glutamate is converted into GABA [27]. However, further investigation is required to uncover the content of L-Glutamate needed to successfully synthesize GABA.

Purine metabolites such as hypoxanthine are important indicators for the progression of chronic illnesses. hypoxanthine synthesis during ATP catabolism was described [28,29]. Barsotti and Ipata demonstrated that during ATP catabolism, various enzymes such as adenosine deaminase catalyze the breakdown of adenosine into hypoxanthine [28]. The current study revealed the presence of adenosine in raw alfalfa seed while its sprouts did not contain the compound. On the contrary, hypoxanthine was only present in the alfalfa sprout extracts. Consequently, we hypothesized that during germination the seedling requires increased energy to break the dormancy state. The increased energy requirements led to an active ATP catabolism through ATP pathways resulting in adenosine being actively converted into hypoxanthine during sprouting [29]. High levels of hypoxanthine in the body are associated with an increased risk of insulin resistance [29]. In vivo study demonstrated that the compound can actively accumulate free radicals in the body, especially nitric oxide (NO_3_) [30]. Thus, hypoxanthine can be considered a biomarker for diabetes, indicating the progression of the condition [29].

Moreover, due to the active deamination process in plants during the early stage of germination, the processes that may lead to the synthesis of organic acids are triggered [24]. The biosynthesis of malic acid during P. mungo seed germination [24] and in animals [31] was described. From Figure 4C, it is noticeable that the malic acid content in sprouts was almost the same as the loss of L-asparagine in alfalfa seeds. Earlier, Morohashi and Shimokoriyama showed that a high amount of malic acid is leached out into the soaking medium during the early period of germination [24]. Thus, there is a strong possibility that active malic acid synthesis took place through deamination of L-Asparagine during the germination process. Since the levels of malic acid were lower in sprouts compared to seeds, it is remarkable that significant leaching of the acid into the soaking water occurred. The malic acid which is leached out is presumed to be the amount that is previously present in the raw seeds while the quantity detected in the sprout is from the metabolic deamination of L-Asparagine [24]. Nevertheless, the understanding of the physiological leaching process of malic acid is still not clear and further research is needed to elucidate this point. The inhibitory effect of malic acid on alpha-glucosidase has been reported [32]. In diabetic control, malic acid can actively be involved in the inhibitory effects of the enzyme as a non-competitive inhibitor and by direct docking to the glucose binding sites [32]. Owing to this fact, consuming malic acid-rich sprouts may enhance the prevention of diabetes. Figure 5A,B describe the mechanism of the formation of GABA and hypoxanthine, respectively.

Total phenolics, flavonoid, and saponins were also investigated in this study. Phenolic compounds are important secondary metabolites in plants because they participate in various processes in reproduction and growth as well as protecting seedlings against environmental stresses and diseases. Table 3 illustrates the effect of germination on the TPC and TFC of alfalfa and buckwheat seeds in the current study. Germination significantly decreased the TPC in both alfalfa and buckwheat seeds (from 425.7 ± 5.1 to 406.1 ± 0.2 mg CE/100 g, DW in alfalfa seeds) and (from 352.1 ± 11.1 to 315.8 ± 4.9 mg CE/100 g in buckwheat seed). The observations are in contrast with what most researchers report, germination improves the TPC of seeds. Zhang et al. reported an increased TPC with an increasing period of germinating buckwheat seeds [34]. Zinca and Vizireanu also noted high levels of TPC in alfalfa seeds at the initial stages of germination but the concentration reduced as the germination process proceeded [4]. The current results were consistent with some reported literature [16]. Beitane and colleagues observed a decrease from 132.85 to 74.13 GAE mg 100 g^−1^ DW of TPC content in buckwheat seeds during germination. The findings of these studies may differ depending on the germination period, seed quality, and environmental conditions under which sprouting is performed [4]. The current findings also confirmed the previous conclusions that buckwheat and alfalfa grains contain a considerable level of flavonoids. Nonetheless, consistent with TPC results, germination significantly reduced the TFC of buckwheat and alfalfa seeds in agreement with the already reported findings [16]. Flavonoids are phenolic compound subclasses and can be affected by similar factors influencing TPC levels. The activation of endogenous enzymes such as guaiacol peroxidase (GPX) during germination may also result in reduced phenol levels in sprouts. GPX catalyzes the conversion of phenolic compounds to polymerized forms such as lignin and lignans (cell wall lignification) to defend seedlings from external stress [35].

Saponins are considered for their ability to reduce obesity-related risks by lowering the cholesterol levels in the blood [36]. However, some of saponins are toxic and regarded as antinutritional constituents [36]. In the current study, a significant (*p* < 0.05) decrease was observed in TSC of buckwheat seeds upon germination (80.8 to 25.4 ± 1.9 mg soy saponin B equivalent/100 g, DW) while a significant increase was revealed in alfalfa seeds (24.5 ± 1.3–41.9 ± 1.6 mg soy saponin B equivalent/100 g, DW) Table 3). Recent findings also reveal that raw buckwheat seeds contain a higher saponin content compared to raw alfalfa seeds. Buckwheat saponins are considered antinutrients [37]. Therefore, the germination process was an effective technique to reduce their concentration to more considerable levels. On the other hand, the alfalfa saponin–cholesterol interactions have been shown as an appropriate mechanism to enhance the hypocholesterolemic activities of the plant in human health [38]. Thus, germination was an appropriate technique to improve the levels of saponin in alfalfa sprouts thereby enhancing their antiobesity activities.

It is widely recognized that plant-based diets rich in antioxidant phytochemicals have the potential to reduce the incidence of various chronic illnesses associated with oxidative stress [4]. Sprouts of alfalfa and buckwheat can offer substantial phytonutrient-rich antioxidant constituents [4,16]. Alfalfa and buckwheat seeds and sprouts were analyzed for their ability to inhibit free radicals by measuring their DPPH and ABTS scavenging activities. The DPPH and ABTS results are shown in Table 3. For the DPPH activity, buckwheat sprout was the most improved by germination, showing an inhibition capacity of 12.20 ± 0.61 µmol TE/g, DW. Alfalfa sprout and raw buckwheat seeds showed µmol TE/g, DW values of 5.23 ± 0.20 and 3.85 ± 0.33, respectively, with no significant difference. Alfalfa seeds had the lowest DDPH value (2.73 ± 0.23 µmol TE/g, DW). Consistent with DPPH findings, buckwheat sprout extracts had the strongest ability to scavenge ABTS radicles (19.73 ± 0.10 µmol TE/g, DW) followed by buckwheat seeds and alfalfa sprouts (15.28 ± 0.43 and 14.85 ± 0.20 µmol TE/g, DW), respectively, while raw alfalfa seeds had the lowest scavenging capacities (12.42 ± 0.18 µmol TE/g, DW). The current findings confirm previous reports that germination can effectively enhance the antioxidant capacities of alfalfa seeds [4] and buckwheat seeds [34] which may be due to the synthesis of compounds with antioxidant potential such as GABA and L-ascorbic acid already discussed in this study.

Furthermore, nowadays, alfalfa and buckwheat sprouts are gaining popularity due to their ability to inhibit digestive enzymes which contribute to obesity and diabetes. So far, studies have reported on changes in the enzyme properties that occur during the germination of buckwheat [39] and alfalfa seeds [2]. In this research, we investigated the alterations in the enzyme activities after the germination of the seeds. The results of enzyme inhibitory activities of 1 mg/mL ethanol extracts were recorded according to Figure 4A,B. Germination led to a sharp increase in the pancreatic lipase inhibitory activities of alfalfa seeds while it did not significantly affect the activities of buckwheat. Alfalfa sprouts recorded lipase inhibitory activities of 57.12 ± 2.0%, which was lower compared to the prescribed drug, orlistat (91.9 ± 1.2%). Alfalfa seeds, buckwheat seeds, and buckwheat sprouts recorded anti-lipase properties of 37.6 ± 3.2, 37.9 ± 2.7, and 36.1 ± 2.5%, respectively, with no significant difference. Moreover, the result of the α-glucosidase activities indicated that the inhibitory capacity of alfalfa seeds was significantly improved by germination from 89.9 ± 0.9 to 96.6 ± 1.0% whereas those of buckwheat seeds was significantly reduced from 96.5 ± 1.1 to 90.1 ± 1.3%. We reported above about an increased saponin content of alfalfa sprouts which may be associated with their increased antiobesity activity. Our observations did not agree with those described by [2] who showed that during germination of alfalfa seeds, α-glucosidase activity decreases. The effectiveness of buckwheat seeds to inhibit the α-glucosidase activities might have diminished upon germination due to the reduced levels of active compounds with enzyme inhibitory activities such as catechin.

The glycation process in diabetic conditions alters the macromolecular structure and body functions, along with causing oxidative stress [3]. The risk factors for AGE formation are diabetes, obesity, and glycation stress, among others [3]. Figure 6 describes the mechanism of AGEs formation and the implications in diabetic conditions. The investigations on the AGE formation inhibitory activity of buckwheat [40] and alfalfa [41] have been reported. The findings of the current research, as shown in Figure 3C, revealed that germination did not significantly affect the AGEs formation activity of buckwheat (27.5 ± 2.2% in seeds and 25.4 ± 3.1% in sprouts) but the process significantly reduced the activity of alfalfa (from 28.7 ± 1.5 to 19.7 ± 2.5%) which may be as a result of a decrease in the levels of key metabolites with AGE properties such as fumaric acids [14].

## 5. Conclusions

The present study explored the effect of sprouting on the antioxidant, antidiabetic, and antiobesity activities in vitro, and the metabolite profiles, of alfalfa and buckwheat. Sprouts demonstrated better antioxidant activities compared to raw seeds. Germination significantly improved the α-glucosidase and pancreatic lipase activities of alfalfa seeds and also enhanced the α-glucosidase activity of buckwheat seeds. Furthermore, the sprouting process improved the overall metabolite profiles of the seeds. The process played a key role in the conversion of L-glutamate into GABA, adenosine into hypoxanthine, and L-asparagine into malic acid. The different metabolite changes observed for buckwheat and alfalfa seeds demonstrate the potential of distinct metabolic switches during the germination process of seeds. Thus, the findings of this research reveal a promising potential of using germination to modify secondary metabolites and health functions of edible seeds. Future research should focus on in vivo analysis of alfalfa and buckwheat seeds and sprouts to substantiate the in vitro findings.

## Figures and Tables

**Figure 1 antioxidants-10-01613-f001:**
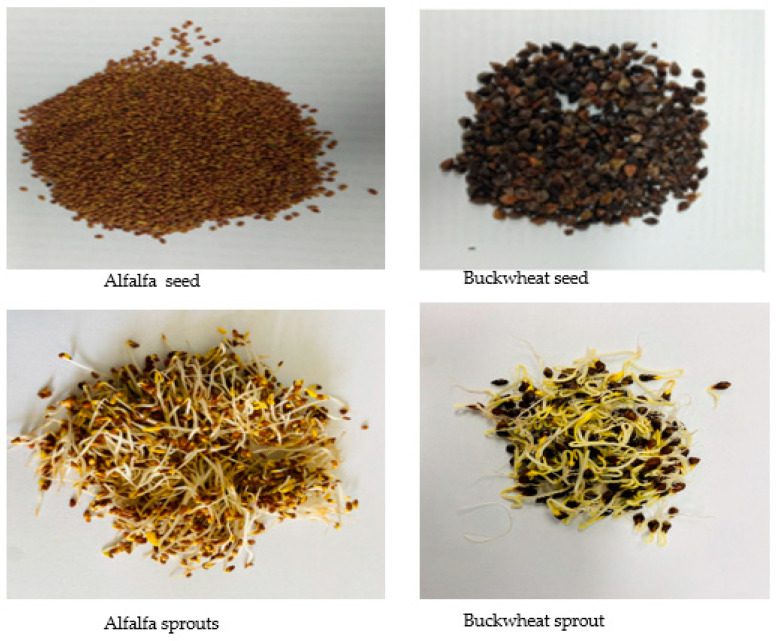
Representatives of alfalfa and buckwheat seeds and sprouts provided by Charm-Chae-One, Ltd. (Jincheon, Chungbuk, Korea).

**Figure 2 antioxidants-10-01613-f002:**
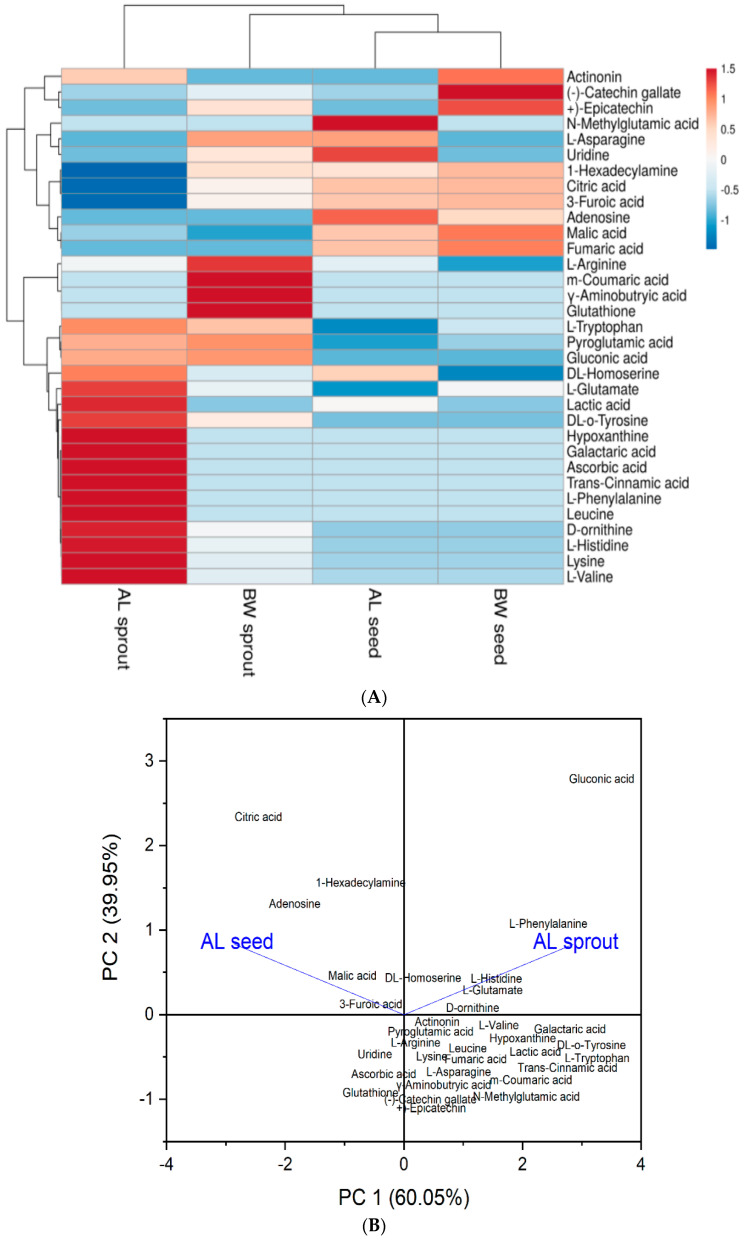
The heat map and biplots result from the heat map and PCA analysis of the metabolites. (**A**) Heat map figure showing the levels of the metabolites in the extracts. (**B**) Biplots of the metabolites identified in alfalfa seed and sprouts. (**C**) Biplots of the metabolites identified in buckwheat seed and sprouts. (**D**) Combined biplots for alfalfa and buckwheat (seed and sprouts). AL, alfalfa; BW, buckwheat.

**Figure 3 antioxidants-10-01613-f003:**
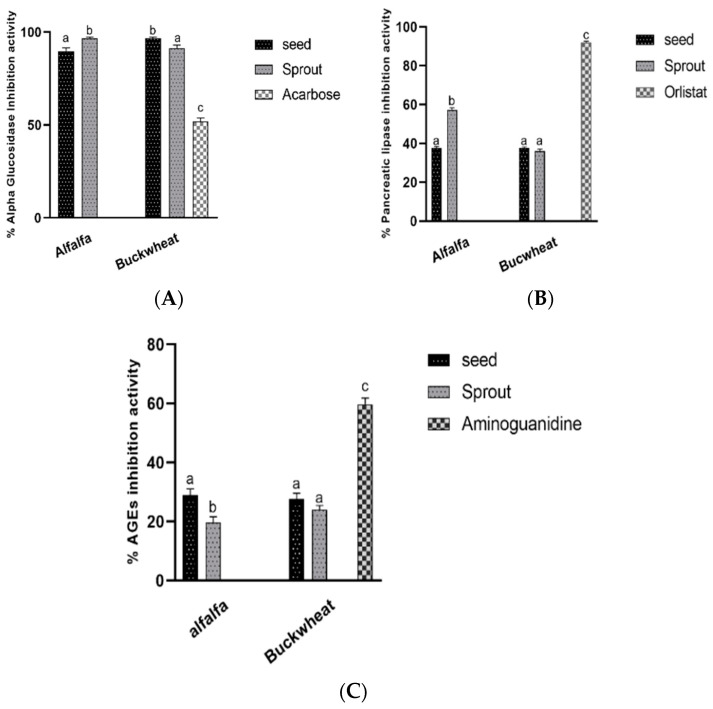
Antidiabetic activity in vitro of mg/mL ethanol extracts (ABCD). (**A**) Percentage α-glucosidase inhibitory activity, (**B**) percentage pancreatic lipase inhibitory activity, (**C**) percentage AGE formation inhibitory activity of alfalfa and buckwheat. Abcd are the lebels of the figures.

**Figure 4 antioxidants-10-01613-f004:**
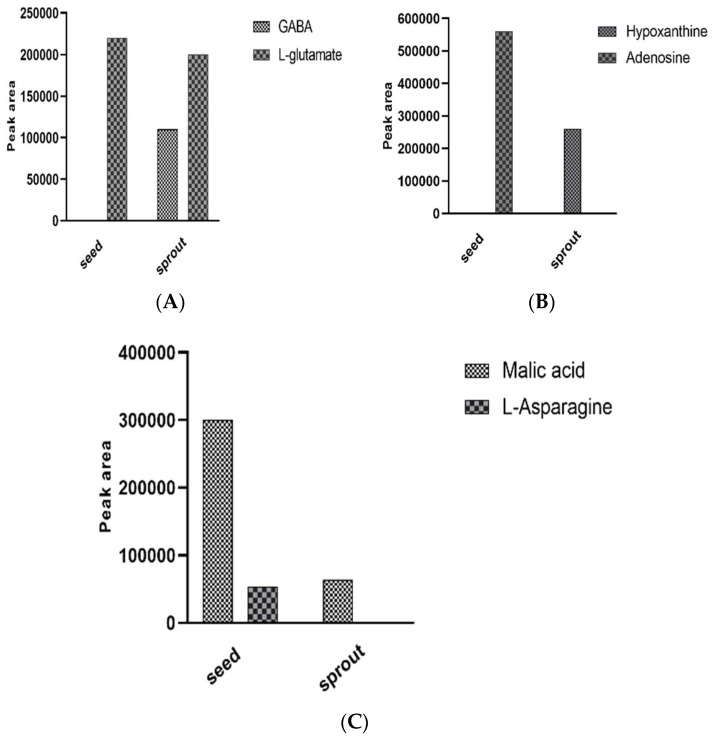
Representatives of bioconverted metabolites. (**A**) GABA, initially lacking in the raw buckwheat seeds, was synthesized from L-glutamate. (**B**) Hypoxanthine, previously not present in raw alfalfa seeds, was synthesized from adenosine. (**C**) Malic acid was synthesized in alfalfa sprouts from L-asparagine in alfalfa seeds.

**Figure 5 antioxidants-10-01613-f005:**
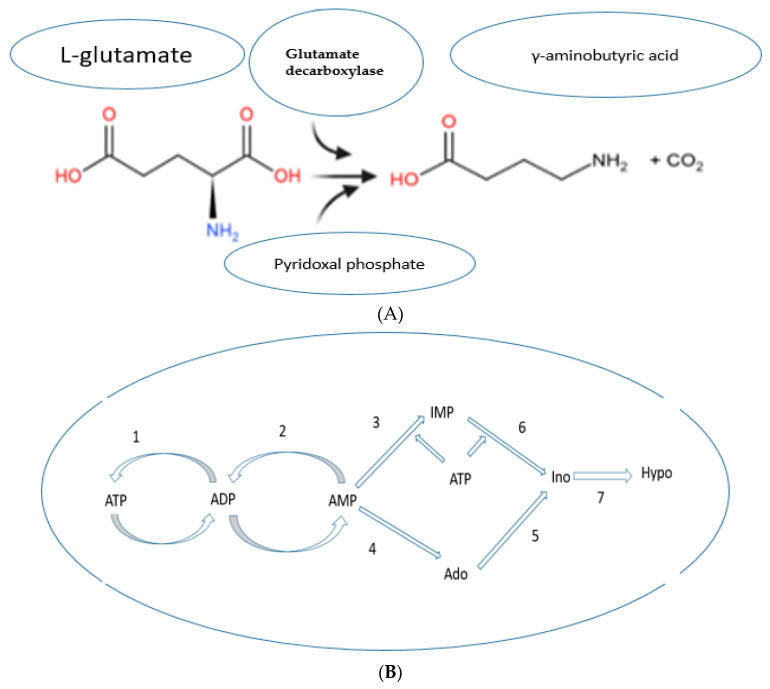
Proposed pathway for the formation of GABA and hypoxanthine. (**A**) Conversion of L-glutamate into γ-Aminobutyric acid (GABA) by decarboxylation process. Obtained from [33]. (**B**) The pathway of intracellular ATP catabolism involved in the synthesis of hypoxanthine [28]. The numbers (1–7) represent various enzyme-catalyzed stages. The ATP is interconverted into ADP and then into AMP which is deaminated into IMP followed by dephosphorylation to produce inosine (Ino). Ino is converted into hypoxanthine. Alternatively, AMP may be dephosphorylated to adenosine (Ado) and then deaminated into inosine which forms hypoxanthine (hypo). IMP, Inosine 5’-monophosphate; GMP, Guanosine monophosphate; ADP, Adenosine diphosphate; ATP, Adenosine triphosphate.

**Figure 6 antioxidants-10-01613-f006:**
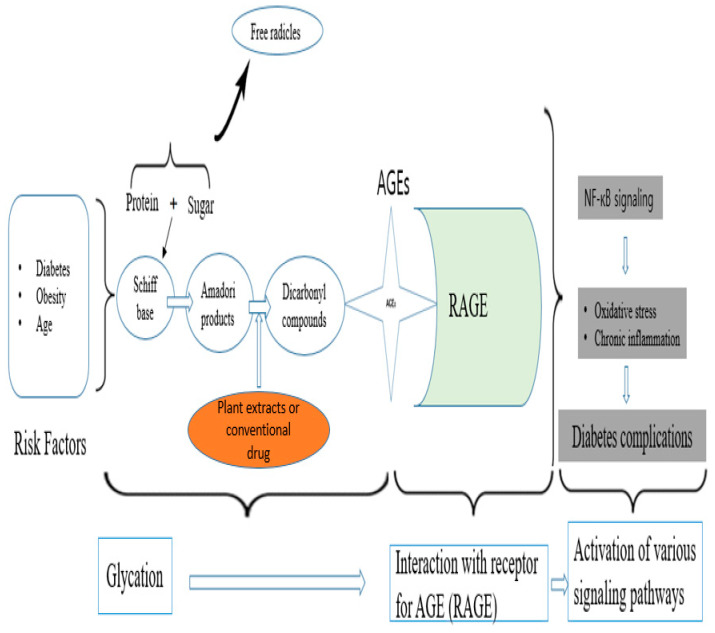
Schematic representation of proposed pathway for formation of advanced glycation end products (AGEs). Early stages of AGE formation involve development of intermediate products. At this stage, free radicles are also released. The intermediate products disintegrate to form AGEs. The plant or conventional drug can inhibit the formation of AGEs at this stage and prevent their progression. The second stage involves interaction of AGE and receptor for AGEs (RAGE), leading to the activation of various genes such as NF-κB target genes involved in inflammation. NF-κB is an inducible transcription factor and when activated it can, in turn, stimulate other transcription genes which regulate inflammation. In the final phase, the inflammation and oxidative stress due to free radicles lead to diabetes complications.

**Table 1 antioxidants-10-01613-t001:** Metabolites that were identified in the ethanol extracts of alfalfa and buckwheat seed extracts by UHPLC-Q-TOF-MS^2^. RT, retention time.

Peak No.	RT Per Min.	Molecular Weight	[M−H]-(*m*/*z*)	Molecular Formula	MS/MS (% Abundance)	Compound Identified
1.	0.70	174.11	173.10	C_6_H_14_N_4_O_2_	131 (100%), 173 (38%)	L-Arginine
2.	0.78	132.05	131.04	C_4_H_8_N_2_O_3_	131 (100%)	L-Asparagine
3.	0.82	147.05	146.04	C_5_H_9_NO_4_	102 (100%), 146 (100%)	L-Glutamate
4.	1.04	116.01	115.00	C_4_H_4_O_4_	115 (100%)	Fumaric acid
5.	1.05	134.02	133.01	C_4_H_6_O_5_	71 (100%), 133 (45%), 59 (30%)	Malic acid
6.	1.22	244.07	243.06	C_9_H_12_N_2_O_6_	243 (100%), 82 (45%)	Uridine
7.	1.22	112.02	111.01	C_5_H_4_O_3_	111(100%), 67 (100%)	3-Furoic acid
8.	1.22	192.03	191.02	C_6_H_8_O_7_	191 ((15%)	Citric acid
9.	1.22	129.04	128.03	C_5_H_7_NO_3_	129 (10%)	Pyroglutamic acid
10.	1.23	267.10	268.10	C_10_H_13_N_5_O_4_	136 (100%), 268 (10%)	Adenosine
11.	1.18	161.07	160.06	C_6_H_11_NO_4_	58 (100%), 142 (50%)	N-Methylglutamic acid
12.	5.95	204.09	203.08	C_11_H_12_N_2_O_2_	116 (40%), 142 (18%), 74 (8%), 186 (6%)	L-Tryptophan
13.	9.68	290.08	289.07	C_15_H_14_O_6_	289 (20%)	(+)-Epicatechin
14.	13.36	385.26	384.25	C_19_H_35_N_3_O_5_	112 (70%), 113 (18%), 224 (15%),	Actinonin
15.	16.24	442.09	441.08	C_22_H_18_O_10_	125 (80%), 124 (15%), 145 (23%), 303 (2%)	(−)-Catechin gallate

**Table 2 antioxidants-10-01613-t002:** Metabolites that were identified in the ethanol extracts of alfalfa and buckwheat sprout extracts by UHPLC-Q-TOF-MS^2^. RT, retention time.

Peak No.	RT Per Min.	Molecular Weight	[M−H]-(*m*/*z*)	Molecular Formula	MS/MS (% Abundance)	Compound Identified
1.	0.69	146.11	145.10	C_6_H_14_N_2_O_2_	145 (100%)	Lysine
2.	0.72	174.11	173.10	C_6_H_14_N_4_O_2_	173 (100%)	L-Arginine
3.	0.72	155.07	154.06	C_6_H_9_N_3_O_2_	93 (100%), 67 (23%), 137 (15%)	L-Histidine
4.	0.79	132.05	131.04	C_4_H_8_N_2_O_3_	131 (100%)	L-Asparagine
5.	0.81	132.09	131.08	C_5_H_12_N_2_O_2_	88 (100%), 131 (52%)	D-ornithine
6.	0.81	119.06	118.05	C_4_H_9_NO_3_	74 (100%) 100 (100%)	DL-Homoserine
7.	0.82	103.06	102.05	C_4_H_9_NO_2_	102 (100%)	γ-Aminobutryic acid
8.	0.82	147.05	146.04	C_5_H_9_NO_4_	102 (39%), 128 (30%)	L-glutamate
9.	0.85	196.06	195.05	C_6_H_12_O_7_	75 (100%), 59 (62%), 85 (37%), 129 (10%), 99 (5%)	Gluconic acid
10.	0.86	210.04	209.03	C_6_H_10_O_8_	71 (100%), 57 (96%), 85 (60%), 129 (15%)	Galactaric acid
11.	0.82	129.04	128.03	C_5_H_7_NO_3_	128 (100%)	Pyroglutamic acid
12.	0.92	176.03	175.02	C_6_H_8_O_6_	129 (100%), 57 (100%)	L-ascorbic acid
13.	1.01	117.08	116.07	C_5_H_11_NO_2_	116 (100%)	L-Valine
14.	1.06	134.02	133.01	C_4_H_6_O_5_	71 (100%), 133 (30%)	Malic acid
15.	1.08	307.08	306.07	C_10_H_17_N_3_O_6_S	143 (100%), 128 (98%), 167 (50%), 74 (49%)	Glutathione
16.	1.21	90.03	89.02	C_3_H_6_O_3_	89 (100%)	Lactic acid
17.	1.23	112.02	111.01	C_5_H_4_O_3_	111 (100%), 67 (100%)	3-Furoic acid
18.	1.23	244.07	243.06	C_9_H_12_N_2_O_6_	243 (80%), 110 (72%), 84 (50%),	Uridine
19.	1.23	136.04	135.03	C_5_H_4_N_4_O	65 (100%), 66 (40%), 106 (9%), 92 (8%)	Hypoxanthine
20.	1.24	181.07	180.06	C_9_H_11_NO_3_	119 (100%), 180 (99%), 93 (45%), 163 (40%)	DL-o-Tyrosine
21.	1.24	192.03	191.02	C_6_H_8_O_7_	191 ((15%)	Citric acid
22.	1.26	131.09	130.08	C_6_H_13_NO_2_	130 (100%)	Leucine
23.	3	165.08	164.07	C_9_H_11_NO_2_	103 (100%), 147 (50%) 77 (50%)	L-Phenylalanine
24.	3	148.05	147.04	C_9_H_8_O_2_	61 (52%)	Trans-Cinnamic acid
25.	5.96	204.09	203.08	C_11_H_12_N_2_O_2_	142 (40%), 74 (28%), 116 (20%)	L-Tryptophan
26.	7.92	164.05	163.04	C_9_H_8_O_3_	91 (50%), 119 (100%), 163 (5%)	m-Coumaric acid
27.	9.68	290.08	289.07	C_15_H_14_O_6_	109 (100%), 137 (83%), 151 (50%)	(+)-Epicatechin
28.	13.36	385.26	384.25	C_19_H_35_N_3_O_5_	224 (39%), 111(37%), 180 (20%)	Actinonin
29.	16.25	442.09	441.08	C_22_H_18_O_10_	125 (80%), 145 (23%), 124 (15%), 303 (2%)	(−)-Catechin gallate
30.	17	241.28	242.29	C_16_H_35_N	136 (100%), 268 (10%)	1-Hexadecylamine

**Table 3 antioxidants-10-01613-t003:** Antioxidant activity (µmol Trolox equivalent/g, DW), total phenolic content (TPC), total flavonoid content (TFC), and total saponin content (TSC) of alfalfa and buckwheat seed and sprouts.

Extracts	TPC (mg Ferulic Acid Equivalent/100 g, DW)	TFC (mg Catechin Equivalent/100 g, DW	TSC (mg Soy Saponin B Equivalent/100 g, DW)	DPPH (µmol Trolox Equivalent/g, DW)	ABTS (µmol Trolox Equivalent/g, DW)
AL seed	425.7 ± 5.1 ^a^	208.4 ± 0.3 ^a^	24.5 ± 1.3 ^a^	2.73 ± 0.22 ^a^	12.42 ± 0.18 ^a^
AL sprout	406.1 ± 0.2 ^b^	198.8 ± 0.4 ^b^	41.9 ± 1.6 ^b^	5.23 ± 0.20 ^b^	14.85 ± 0.20 ^b^
BW Seed	352.1 ± 11.1 ^c^	220.7 ± 0.8 ^c^	80.8 ± 5.5 ^c^	3.85± 0.33 ^b^	15.28 ± 0.43 ^b^
BW sprout	315.8 ± 4.9 ^d^	208.1 ± 0.9 ^a^	25.4 ± 1.9 ^a^	12.20 ± 0.61 ^c^	19.73 ± 0.10 ^c^

Results are expressed as means ± SD. Different superscripts (a–d) within each column denote significant differences (*p* < 0.05). DW, dry weight; AL, alfalfa; BW, Buckwheat.

## Data Availability

Data is contained within the article and Appendix A.

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
