# Peer review of "Effect of Germination on Alfalfa and Buckwheat: Phytochemical Profiling by UHPLC-ESI-QTOF-MS/MS, Bioactive Compounds, and In-Vitro Studies of Their Diabetes and Obesity-Related Functions"

_antioxidants, 2021, doi:10.3390/antiox10101613_

Round 1
Reviewer 1 Report
An intake of a sufficient amount and spectrum of biologically active substances in food or food supplements is a crucial natural way to prevent many diseases of civilization. It is known, especially in the case of cereals, that the germination process in principle increases the content of antioxidants and other physiologically active substances in the grain.
I have a fundamental comment on the article and the presented results. Seeds (non-germinated, it is not mentioned that the obtained seeds were germinated) and buckwheat and alfalfa sprouts were analyzed and compared.
The results and discussions are formulated as a comparison of the profile of analytes or antioxidant activity caused by germination (formulations such as: "germination process was effective in enhancing the health function with respect to antioxidant activity of the seeds").
After water uptake by a seed, processes such as activation or de novo formation of cytolytic, proteolytic, saccharolytic and other enzymes are activated in the germination process, leading to cell structures modification of a seed and the formation of metabolites necessary for the growth of the sprouts and roots of the new plant. The composition of a "body" of germinating seed and germ is different. Thus, the analytical profile of non-germinated grain versus the analytical profile of sprouts should be discussed, and the title of the article should be changed.
Author Response
Dear Editor,
RESPONSE TO REVIEWER COMMENTS
We are grateful for your valuable comments and we have carefully revised the manuscript as the reviewers suggested. Please find the response to the reviewer comments. Comments are in red and responses in black.
Regards,
Deog-Hwan Oh (PhD)
Reviewer #1
Comment: I have a fundamental comment on the article and the presented results. Seeds (non-germinated, it is not mentioned that the obtained seeds were germinated) and buckwheat and alfalfa sprouts were analyzed and compared.
Response: the seeds were obtained from the vendor who stated the days and the conditions of germination. This information is mentioned in line 65-66
Comment: After water uptake by a seed, processes such as activation or de novo formation of cytolytic, proteolytic, saccharolytic and other enzymes are activated in the germination process, leading to cell structures modification of a seed and the formation of metabolites necessary for the growth of the sprouts and roots of the new plant. The composition of a "body" of germinating seed and germ is different. Thus, the analytical profile of non-germinated grain versus the analytical profile of sprouts should be discussed, and the title of the article should be changed.
Response: the profiles of alfalfa and buckwheat (seeds and sprouts) were extensively assessed and discussed (table 1 and table 2 provides the metabolite profiles from these samples, which have been subsequently discussed from line 268-349). The metabolites were further analyzed as shown on the heatmap (figure 2a) and PCA bi-plot figures (2b-d)
Comment: The title of the article should be changed
Response: the manuscript title was changed as “Effect of Germination on Alfalfa and Buckwheat: Phytochemical profiling by UHPLC-ESI-QTOF-MS/MS, bioactive compounds, and in vitro Studies of their Diabetes and Obesity-related Functions”

Reviewer 2 Report
The manuscript entitled “Profiling and Establishing Health-Beneficial Effects of Bioactive Compounds in Germinated Alfalfa and Buckwheat Seeds” and authored by Simon Okomo Aloo , Fred Kwame Ofosu , Deog-Hwan Oh, deals with the investigation of the effect of sprouting on the antioxidant, antidiabetic, anti-obesity activities and metabolite profiles of alfalfa and buckwheat seeds.
The article contains really interesting data, which deserve to be published in a prestigious journal as Antioxidant. Furthermore, the employed methodologies, although not innovative, were used with criteria and rationality. The results section is well descriptive, and the Discussion is supported by valid bibliographic sources. Maybe, the authors could update some of the references as some turn out to be truly dated.
Keywords should be words not contained in the title, at most present in the abstract. Their usefulness is to make easier the searching of the article using the common scientific search engines. Since several keywords are already present in the title, and/or repeated several times in the abstract, I strongly advise the authors to replace some of them and add more. As journal guidelines clearly report, a limited number of keywords can be used (maximum 10). Consequently, authors should carefully choose them.
In the revised manuscript, the authors performed mass-spectrometric analyzes using UHPLC-QTOF-MS / MS. However, this important aspect is not highlighted neither in the title, nor in the abstract, nor in the keywords of the manuscript. I would strongly suggest to the authors to introduce some information at least in the abstract, and a keyword in the appropriate sections,
The first panel of Figure 1 appears to have a low resolution. Moreover, the text boxes of the upper panels cut out the figures of the lower panels. Authors should fix these problems and increase the quality of the figure. Finally, as reported in the journal's guidelines, each panel should be designed with a capital letter (A, B, C, D), placed at the top left of each panel, in UPPERCASE and bold. Finally, the description of each panel should be given in the legend of the figure.
The chromatographic conditions (column temperature, elution gradient, etc.), along with the analysis parameters of the mass spectrometer are completely missing in the Materials and Methods section. Please, complete 2.3.10 subsection.
Please, move line 183-185 into the results section.
In the main text, several typos are present. For example MS / MS2, g-1, [M-H] -, etc. Authors should carefully check the manuscript and fix them.
Author Response
Dear Editor,
RESPONSE TO REVIEWER COMMENTS
We are grateful for your valuable comments and we have carefully revised the manuscript as the reviewers suggested. Please find the response to the reviewer comments. Comments are in red and responses in black.
Regards,
Deog-Hwan Oh (PhD)
Reviewer #2
Comment: Keywords should be words not contained in the title, at most present in the abstract. Their usefulness is to make easier the searching of the article using the common scientific search engines. Since several keywords are already present in the title, and/or repeated several times in the abstract, I strongly advise the authors to replace some of them and add more. As journal guidelines clearly report, a limited number of keywords can be used (maximum 10). Consequently, authors should carefully choose them.
Response: The key words have been replaced as “Metabolites; seeds; sprouts; antioxidant; enzyme inhibition”
Comment: In the revised manuscript, the authors performed mass-spectrometric analyzes using UHPLC-QTOF-MS / MS. However, this important aspect is not highlighted neither in the title, nor in the abstract, nor in the keywords of the manuscript. I would strongly suggest to the authors to introduce some information at least in the abstract, and a keyword in the appropriate sections,
Response: UHPLC-QTOF-MS / MS has been mentioned in the title and added in key words (line 3 and line 25)
Comment: The first panel of Figure 1 appears to have a low resolution. Moreover, the text boxes of the upper panels cut out the figures of the lower panels. Authors should fix these problems and increase the quality of the figure.
Response: The figure resolution was improved and the text boxes fixed (line 70)
Comment: Finally, as reported in the journal's guidelines, each panel should be designed with a capital letter (A, B, C, D), placed at the top left of each panel, in UPPERCASE and bold. Finally, the description of each panel should be given in the legend of the figure.
Response: All the figures were renamed with capital letters (ABCD) in the entire manuscript
Comment: The chromatographic conditions (column temperature, elution gradient, etc.), along with the analysis parameters of the mass spectrometer are completely missing in the Materials and Methods section. Please, complete 2.3.10 subsection.
Response: the following information was added to the 2.3.10 subsection “The Q-TOF-MS was set for the negative mode in a mass range of 100–1000 and a resolution of 5000. The capillary voltage was 1.45 kV while cone voltages used was 30 V. The flow rate of Helium (gas in the cone) was 45 L/h, and the flow rate of desolvation gas (nitrogen gas, N2) was 900 L/h. The temperature of nitrogen gas was about 250 °C, and the ion source temperature was 120 °C. The collision energies needed to record the MS/MS spectra were established at 15, 20, and 30 V” line 203-208)
Comment: Please, move line 183-185 into the results section.
Response: the line 183-185 was moved to 196-201)
Comment: In the main text, several typos are present. For example, MS / MS2, g-1, [M-H] -, etc. Authors should carefully check the manuscript and fix them.
Response: the typing errors were corrected in lines 203, 219, 222, 290, 440

This manuscript is a resubmission of an earlier submission. The following is a list of the peer review reports and author responses from that submission.
Round 1
Reviewer 1 Report
The context should be better introduced;particularly the authors should better describe the profile of bioactive components of buckwheat and related references added such as:
Durazzo et al.Lignan content in cereals, buckwheat and derived foods. Foods, 2013,2,53-63.
Kreft et al. 2016.Buckwheat polyphenols metabolites in health and diseases.Nutr.Res.Rev.2016
A graphical scheme of approach of study should be inserted.
Introductory lines in section Results should be inserted.
Subparagraph 3.1 Metabolite identification should be implemented
Data in Figures 2 should be better described in the text.
Major details should be inserted in Figure 5
Limits, advantages and novelty of paper should be stated